# Comparison of the Efficacy and Safety of Adamgammadex with Sugammadex for Reversal of Rocuronium-Induced Neuromuscular Block: Results of a Phase II Clinical Trial

**DOI:** 10.3390/jcm11236951

**Published:** 2022-11-25

**Authors:** Yingying Jiang, Yujun Zhang, Zhaoqiong Zhu, Yidan Huang, Dachun Zhou, Jingchen Liu, Chaoyu Li, Xiangdong Chen, Dingxin Kang, Shoushi Wang, Jin Liu, Bin Liu, Wensheng Zhang

**Affiliations:** 1Department of Anesthesiology, West China Hospital, Sichuan University, Chengdu 610041, China; 2National-Local Joint Engineering Research Center of Translational Medicine of Anesthesiology, West China Hospital, Sichuan University, Chengdu 610041, China; 3Laboratory of Anaesthesia and Critical Care Medicine, Translational Neuroscience Center, West China Hospital, Sichuan University, Chengdu 610041, China; 4Department of Anesthesiology, The First Affiliated Hospital of Zunyi Medical University, Zunyi 563000, China; 5Department of Anesthesiology, Liuzhou People’s Hospital, Liuzhou 545000, China; 6Department of Anesthesiology, Zhejiang University School of Medicine Sir Run Run Shaw Hospital, Hangzhou 310000, China; 7Department of Anesthesiology, The First Affiliated Hospital of Guangxi Medical University, Nanning 530021, China; 8Department of Anesthesiology, The Second People’s Hospital of Neijiang, Neijiang 641000, China; 9Department of Anesthesiology, Union Hospital, Tongji Medical College, Huazhong University of Science and Technology, Wuhan 430022, China; 10Department of Anesthesiology, The 2th Affiliated Hospital of WMU, Wenzhou 325000, China; 11Department of Anesthesiology, Qingdao Central Hospital Group, Qingdao 266000, China

**Keywords:** perioperative medicine, muscle relaxant antagonist, phase II clinical trial, pharmacodynamics, adamgammadex, sugammadex

## Abstract

This current phase II clinical trial was to compare the effect and safety of adamgammadex, a new cyclodextrin-based selective relaxant binding agent, with sugammadex to reverse rocuronium-induced neuromuscular block. Patients were randomised to receive adamgammadex (4 or 6 mg kg^−1^) or sugammadex (2 mg kg^−1^, as a positive control group) at the reappearance of the second twitch (T_2_) in response to TOF stimulation. The standard safety data were collected. The 4 mg kg^−1^ (*n* = 16) and 6 mg kg^−1^ (*n* = 20) adamgammadex- and 2 mg kg^−1^ (*n* = 20) sugammadex-induced recovery time of TOF ratio to 0.9 were 2.3, 1.6, and 1.5 min, respectively (*p* = 0.49). The 4 mg kg ^−1^ adamgammadex-induced median recovery time was longer than that of 2 mg kg^−1^ sugammadex (*p* = 0.01), and there was no difference between the 6 mg kg ^−1^ adamgammadex group and 2 mg kg^−1^ sugammadex group (*p* = 0.32). Then, the number of patients who experienced adverse events (AEs) was 6, 11, and 14 for adamgammadex at 4, 6 mg kg^−1^ and sugammadex at 2 mg kg^−1^, respectively. The treatment emergent AEs that occurred more than twice were detailed as follows: incision site pain, hypotension, emesis, fever, throat pain, blood bilirubin increase, abnormal T-wave of ECG, dizziness, incision site swelling, postoperative fever, expectoration, and nausea. For drug-related AEs, the increased urine acetone bodies and first-degree atrioventricular block were observed in two patients from sugammadex group. Then, the previously reported AEs were not observed in this study, including anaphylaxis, haemorrhage, recurarization, abnormal basic vital signs, or lengthened QRS intervals and QT intervals. Adamgammadex was found to be effective for reversal of rocuronium-induced neuromuscular block as sugammadex.

## 1. Introduction

Sugammadex, the first cyclodextrin-based selective relaxant binding agent, has been approved for clinical use in seventy countries around the world [1]. Despite the many advantages of sugammadex, there are concerns about its potential side effects. Previous studies demonstrated that sugammadex could increase the risk of postoperative bleeding, bradycardia, recurarization, hypersensitivity reactions, and anaphylaxis [2,3]. These potential side effects possibly related to sugammadex are rare; sugammadex-induced anaphylaxis was reportedly calculated as occurring in 1 in 2500 administrations (0.039%) [4]. However, it is worth noting that the FDA delayed the approval of sugammadex in the United States several times largely predicated on the concerns surrounding hypersensitivity reactions. Thus, developing a novel cyclodextrin-based selective relaxant binding agent with lower side effects is warranted for the clinical use.

Relevant studies of penicillin-induced anaphylaxis demonstrated that the beta-lactam carboxyl group on the side chains of penicillin is an important antigenic determinant and that allergic cross-reactions occurs in drugs with similar structure [5]. Hence, the risks of allergy to sugammadex, and even other side effects, may result from the reactions between the carboxyl groups of sugammadex with some of the tissues or substances, such as intrinsic proteins. Thus, adamgammadex sodium (Figure 1), designed and synthesized by Hangzhou Adamerck Pharmlabs Inc., has the same nuclear structure (γ-cyclodextrin) as sugammadex; however, the α-carbon of carboxylic acid of adamgammadex was changed to chiral acetylamino. More chiral carbon atoms to the nucleus of γ-cyclodextrin may reinforce the chiral environment and increase the steric hindrance of the carboxyl groups, thereby strengthening its ability to bind rocuronium and reducing the risk of anaphylaxis induced by adamgammadex [6]. However, according to the methods of isothermal titration calorimetry, ultraviolet-visible spectrophotometry and nuclear magnetic resonance spectroscopy, the binding ability of adamgammadex with rocuronium and vecuronium is stronger than that of sugammadex and γ-cyclodextrin [7]. Meanwhile, a preclinical study demonstrated that adamgammadex could restore the effect of rocuronium in a dose-dependent relationship in beagle dogs, and the risk of potential side effects, including hypersensitivity, bleeding, and cardiovascular toxicity, was lower than that of sugammadex in zebrafish [6]. In a phase I study [8], eight healthy volunteers tolerated adamgammadex administered at a dose of 32 mg kg^−1^ well; no serious adverse effects (AEs) were observed, and eleven AEs in the eight participants were judged to be possibly related to adamgammadex: seven cases of increased serum ketone body (16% of the participants), two cases of increased urinary glucose (5% of the participants), and two cases of increased urinary ketone body (5% of the participants). The incidence rate of adamgammadex-induced AEs was lower than that of the saline-induced AEs. A pharmacokinetic study demonstrated that adamgammadex has linear pharmacokinetic properties and that the primary excretion is through the kidneys. In a dose-finding study [9], the time of it took for the train-of-four (TOF) ratio to return to 0.9 in the group that received saline decreased from 39.3 min to within 3 min in the adamgammadex group. In a phase IIa study [10], compared with sugammadex, a similar recovery time of rocuronium-induced deep neuromuscular block was observed in patients administered adamgammadex. Overall, adamgammadex was found to be effective for the reversal of a rocuronium-induced neuromuscular block.

This multi-centre, randomised, double-blind, positive-controlled, phase II clinical trial compared the effect and safety of adamgammadex with sugammadex in patients under a rocuronium-induced moderate neuromuscular block.

## 2. Materials and Methods

### 2.1. Study Design, Ethics, and Criteria of Inclusion and Exclusion

This Phase IIb study (2018-Clinical Trial-58) was approved by the Clinical Trial Ethics Committee of West China Hospital, Sichuan University, and was registered prior to patient enrolment at the Chinese Clinical Trial Registry (ChiCTR2000033548, principal investigator: Jin Liu and Yingying Jiang, Date of registration: 5 June 2020, http://www.chictr.org.cn/showprojen.aspx?proj=48411). The Declaration of Helsinki and the guidance of ICH Good Clinical Practice E6 were used as the guidelines for the formulation of this study protocol (HX-IRB-AF-12-V5.0). The objectives, procedures, and risks of this clinical trial were elaborated to all the participants, and the written informed consent was obtained for all participants. The criteria of inclusion and exclusion were detailed in our previous study [9]. Male and female patients, aged 18 to 64 years old and with ASA physical status at grade 1 or 2, were included if rocuronium was necessary during surgical procedures. Briefly, the exclusion criteria were as follows: difficult airway; neuromuscular diseases; hypertension or hypotension; cardiac, hepatic, or renal function insufficiency; coagulation disorders, malignant hyperthermia syndrome; allergy to cyclodextrin, anaesthetics, muscle relaxants, electrode gels or anything possibly used during surgery; use of prohibited drugs, namely steroidal antibiotics (10 days), progesterone, norethindrone testosterone, prednisone and hydrocortisone (10 days), clomifene citrate, toremifene citrate, tamoxifen citrate, and betamethasone (3 months), and any drugs known to interfere with neuromuscular blockers (such as anticonvulsants, aminoglycosides, and magnesium); participation in any other clinical trial within the past 3 months.

### 2.2. Study Procedures

Patients were randomised sequentially into different groups according to the random number (*n* = 20 in each dose group). The doses of adamgammadex were 4 and 6 mg kg^−1^ according to previous studies [7,9,10]. The positive control, sugammadex, was dosed at 2 mg kg^−1^, based on the dosing information published by the Merck & Co., Inc. (Rahway, NJ, USA). Patients were monitored by various methods after arriving at the operating room, including electrocardiogram (ECG), non-invasive blood pressure, and pulse-oximetry (SpO_2_). The opioid analgesics (fentanyl, remifentanil, or sufentanil) and propofol were used for pain control and sedation depending on the experience of the researcher. Rocuronium at a dose of 0.6 mg kg^−1^ was administered for less than 10 s for tracheal intubation. The ulnar nerve stimulation and the response of adductor pollicis muscle measurement, regarded as the neuromuscular function, were monitored by TOF-Watch^®^ SX (Organon & Co., Dublin, Ireland) [11]. During the surgery, rocuronium at dose of 0.1–0.2 mg kg^−1^ was administered when the T_1_ (first response) or T_2_ (second response) of TOF occurred. When the surgery finished, adamgammadex or sugammadex was injected by the researcher when the reappearance of T_2_. The criterion for adequate neuromuscular recovery occurred when the TOF ratio (TOF_ratio_), the fourth twitch (T_4_) compared to the first twitch (T_1_), had returned to more than 0.9 [12]. No other reversal agents for neuromuscular block were used. Sedation and pain control were suspended after TOF_ratio_ recovered to 0.9. Thereafter, neuromuscular function was monitored until 30 min after neuromuscular recovery to evaluate the recurarization, and the SpO_2_ was further monitored until 1 h after neuromuscular recovery to estimate the recurarization or residual curarization.

### 2.3. Efficacy Study

The primary outcome of this study was the recovery time from the beginning of injection of adamgammadex or sugammadex to the time the TOF_ratio_ returned to 0.9.

### 2.4. Safety Study

All patients were closely observed from the day of the screening visit to the seventh day after drug administration, and safety assessments were carried out by an assessor who was blinded to the medication administered. Basic vital signs, including non-invasive blood pressure, heart rate, and SpO_2_, were measured at the screening visit, before rocuronium administration (regarded as 0 min), and 2, 4, 5, 10, and 30 min after injection of adamgammadex or sugammadex. Twelve-lead ECG was recorded at the following time points: at the screening visit and at 2–4 and 4–8 h after the administration of adamgammadex or sugammadex. Residual neuromuscular block, regarded as recurarization, was observed to have a TOF_ratio_ < 0.9 for more than 30 min, which was regarded as the TOF_ratio_ returned to 0.8 (three times consecutively). As described in a previous study [9], possible symptoms related to anaphylactic reaction, including fever, hypotension, erythema, urticaria, angioedema, bronchospasm, pustules, lymphadenectasis, liver function disorder, eosinophilia, or neutrophilic leucocytosis, were observed, and tryptase level was measured within 3 h once anaphylactic reaction occurred.

Clinical symptoms, anaphylaxis, physical examinations, basic vital signs, ECG, laboratory examinations (haematological parameters, biochemical parameters, coagulation function, urinary parameters), and tryptase (only in the event of anaphylactic reaction) were assessed at the 24 h after adamgammadex or sugammadex administration.

AEs were categorized into five grades according to the Common Terminology Criteria for AEs (CTCAE) version 4.03: grade 1 (mild; asymptomatic or mild symptoms; clinical or diagnostic observations only; intervention not indicated); grade 2 (moderate; minimal, minimal, local or non-invasive intervention indicated; limiting age-appropriate instrumental activities of daily living); grade 3 (severe or medically significant but not immediately life-threatening; hospitalization or prolongation of hospitalization indicated; disabling; limiting self-care activities of daily living); grade 4 (life-threatening consequences; urgent intervention indicated.); and grade 5 (death related to AEs). Grades 4 and 5 were classified as serious adverse events (SAEs).

### 2.5. Statistical Analysis

The data of all subjects who received adamgammadex or sugammadex and had at least one post-dose efficacy measurement were categorized as the intent-to-treat population (full analysis set). The data of members of the intent-to-treat group who had no major protocol violation were categorized as the per-protocol population (per-protocol set). The data of all participants who received adamgammadex or sugammadex were categorized as the safety population (safety set). Statistical analyses were performed by Mosim Co., Ltd. (Shanghai, China) using statistical analysis system software (SAS, version 9.4, Cary, NC, USA). The efficacy was assessed by one-way analysis of variance (ANOVA). The level of statistical significance was set at *p* < 0.05.

## 3. Results

### 3.1. Patients

A total of 63 patients were screened (Figure 2). Then, 60 patients (full analysis set) were averagely randomised into three groups (*n* = 20 in each group), which included 24 male and 36 female patients at aged 20 to 60 years old with body mass index (BMI) between 19 and 30 kg/m^2^ (Table 1). Eighteen patients were ASA grade of 1, while forty-two patients were ASA grade of 2. There was no significant difference in baseline characteristics among the three groups. Four patients from the 4 mg kg^−1^ adamgammadex group declined to participate before drug administration: one patient refused the operative treatment; one patient withdrew consent; and the TOF value of two patients was interrupted due to technical problems with the muscle relaxation monitor. Thus, the data from 56 patients were categorized as the safety set (SS), and the data of 53 patients were categorized as per-protocol set (PPS).

### 3.2. Efficacy Study

The median time of the TOF_ratio_ recovery to 0.9 was within 2.5 min (Figure 3). The median recovery times of the TOF_ratio_ to 0.9 induced by 4 and 6 mg kg ^−1^ adamgammadex- and 2 mg kg^−1^ sugammadex were 2.3, 1.6, and 1.5 min, respectively (*p* = 0.49). Then, we further compared the difference between groups. The 4 mg kg ^−1^ adamgammadex-induced median recovery time was longer than that of 2 mg kg^−1^ sugammadex (*p* = 0.01), and there was no difference between the 6 mg kg ^−1^ adamgammadex group and 2 mg kg^−1^ sugammadex group (*p* = 0.32).

### 3.3. Safety Study

The AE information is presented in Table 2. Only one patient experienced one pre-treatment event (PTE), who was a 25-year-old female patient undergoing laparoscopic surgery for ovarian teratoma. During the screening visit, decreased serum potassium (3.18) was observed and regarded as grade 1, from which she recovered without any treatment. As presented in Table 2, for the treatment-emergent adverse events (TEAEs), the number of patients who experienced more than one AE were 6 (38%), 11 (55%), and 14 (70%) for the groups that administered 4 or 6 mg kg^−1^ doses of adamgammadex, and 2 mg kg^−1^ dose of sugammadex, respectively. The ratio of patients from the 2 mg kg^−1^ sugammadex group who experienced TEAEs was 70%, which was greater than that of the 4 mg kg^−1^ (38%) and 6 mg kg^−1^ (55%) adamgammadex groups, respectively. The TEAEs that occurred more than twice were detailed as follows (Figure 4): incision site pain (20%), hypotension (13%), emesis (11%), fever (9%), throat pain (7%), blood bilirubin increase (7%), abnormal T-wave of ECG (5%), dizziness (4%), incision site swelling (4%), postoperative fever (4%), expectoration (4%), and nausea (4%). For drug-related AEs, the increased urine acetone bodies and first-degree atrioventricular block were observed in two patients from the 2 mg kg^−1^ sugammadex group. According to CTCAE guidance, the grade of those drug-related AEs was 1, and they recovered back to normal without any treatment. Thus, anaphylactic reactions and coagulation function were closely monitored during the clinical trial. However, no clinical symptoms related to anaphylactic reactions occurred. In addition, no cases of postoperative bleeding were observed, and the results of the laboratory examinations of coagulation function were within the normal range (Table 3). Furthermore, the evaluation of basic vital signs and 12-lead ECG are presented in Figure 5 and Figure 6, and there was no significant difference between the adamgammadex and sugammadex groups. No cases of recurarization were observed among the groups.

## 4. Discussion

Similar to sugammadex [13], the selective relaxant binding agent adamgammadex was developed to remove rocuronium from the neuromuscular junction to reverse neuromuscular block. This phase IIb clinical trial was to compare the effect and safety of adamgammadex with sugammadex in patients under the rocuronium-induced moderate neuromuscular block. The results of this study showed that the median recovery time of the TOF_ratio_ to 0.9 by the 6 mg kg^−1^ adamgammadex and 2 mg kg^−1^ sugammadex groups was not statistically significantly different. The incidence of TEAEs in the sugammadex group was greater than that in the adamgammadex groups, and the drug-related AEs were only observed in the sugammadex group. Furthermore, the previously reported AEs, including allergy, post-surgery haemorrhage, recurarization, abnormal basic vital signs, or QRS intervals and QT intervals prolongation, were not observed in this study.

Based on a previous study [14], the advised dose of sugammadex for moderate neuromuscular block is 2 mg kg^−1^ and for deep neuromuscular block is 4 mg kg^−1^. This study compared the efficacy of adamgammadex and sugammadex for moderate neuromuscular block, and the dose choice was an important factor. The recovery time of the TOF_ratio_ to 0.9 induced by adamgammadex at a dose of 4 mg kg^−1^ was significantly longer than that of sugammadex for rocuronium-induced deep neuromuscular block [10], and adamgammadex had a similar effect at a dose of 6 mg kg^−1^. The reason may be related to the chemical structure [7]. The nuclei in the structures of adamgammadex and sugammadex are the same, and the introduction of a chiral acetylamino group to the α-carbon next to the carboxylic acid of each side chain of adamgammadex increases the molecular weight and influences the effect of adamgammadex to some extent. However, the results of the isothermal titration calorimetry tests demonstrated that adamgammadex and sugammadex combined with rocuronium in an isometric molar ratio [7]. Thus, adamgammadex requires a higher dose than sugammadex to produce a similar effect. Additionally, adamgammadex showed a linear pharmacokinetic property at doses from 2 to 32 mg kg^−1^ [8]. Thus, adamgammadex at doses of 4 and 6 mg kg^−1^ was compared with the 2 mg kg^−1^ sugammadex dose for the reversal of rocuronium-induced moderate neuromuscular block.

Sample size calculation, which is an integral component of randomized clinical trials and evaluation of new treatment outcomes, is necessary to assess how many participants are required for the study. Adamgammadex sodium, a class 1.1 new drug, was approved for clinical trials. According to the investigational new drug process published by FDA [15], our current study was phase IIb clinical trial for the comparison of the effect and safety of adamgammadex with sugammadex in patients, and the purpose of phase II studies is to provide preliminary data on the effectiveness of the drug and permit the selection of an appropriate dose range for evaluation in phase III studies. Meanwhile, based on the guidance published by European Medicines Agency (ICH Topic E9 Statistical Principles for Clinical Trials, CPMP/ICH/363/96), the objectives of exploratory trials may not always point to simple tests of pre-defined hypotheses. In addition, exploratory trials may sometimes require a more flexible approach in design to enable changes in response to accumulating results. Thus, the sample size of this study was 20 in each dose group, which was determined based on the phase II clinical trials of sugammadex having sample sizes ranging from 3 [16] to 18 [17] in each dose group. Thus, we considered that the sample size of our study to be reasonable.

The main adverse events related to sugammadex use included three aspects. (1) Anaphylaxis and hypersensitivity: Previous clinical studies demonstrated that the incidence of hypersensitivity was 5% and the incidence of anaphylaxis was 0.3% after any dose of sugammadex [18]. Similar to the observation of anaphylaxis and hypersensitivity [19], the clinical symptoms of anaphylaxis and any other side effects were recorded in this study. None of patients among the groups were evaluated as having an anaphylactic reaction. (2) QT interval prolongation and cardiac arrhythmias: Data from clinical trials showed that sugammadex did not produce clinically relevant QTc prolongation [20,21,22]. In this study, we also demonstrated that the values of QT interval and QTc interval in this study showed no clinical changes after adamgammadex or sugammadex administration. However, the results from other studies showed that the patients had mildly higher QTc interval values after sugammadex administration [23,24]. These data were likely related to the other anaesthetic agents, because those drugs could produce QT interval changes by themselves. Another important adverse effect, cardiac arrhythmias, needs to be considered. Sugammadex induced a variety of cardiac arrhythmias, including bradycardia, atrial or ventricular fibrillation, and atrioventricular and ST segment changes [25]. In this study, only one case (5%) of first-degree atrioventricular block was considered to be induced by 2 mg kg^−1^ sugammadex. In addition, sinus bradycardia was observed in one patient (17%) administered 10 mg kg^−1^ adamgammadex, which was resolved by atropine treatment [10], and phase I clinical trial of adamgammadex also showed no changes in clinically meaningful ECG abnormalities [8]. The inevitable limitation of this phase II clinical trial was the small sample size, and further research is required to better evaluate the safety of adamgammadex. (3) Anticoagulant effects: Sugammadex prolonged the activated partial thromboplastin time and prothrombin time with no significant bleeding excess in clinical practice, which was considered to be induced by binding to phospholipids [26,27,28]. The parameters of coagulation function, including prothrombin time, activated partial thromboplastin time, thrombin time, fibrinogen and international normalized ratio, were measured at 24 h postadministration and were within the normal range.

## 5. Conclusions

The median time of the TOF_ratio_ recovery to 0.9 in the 6 mg kg^−1^ adamgammadex groups showed no significant difference compared with the 2 mg kg^−1^ sugammadex group in patients under rocuronium-induced moderate neuromuscular block. Compared with sugammadex, adamgammadex had a lower incidence of TEAEs, and no drug-related AEs were observed. However, a larger patient cohort trial should be performed to further evaluate the safety and tolerability of adamgammadex.

## Figures and Tables

**Figure 1 jcm-11-06951-f001:**
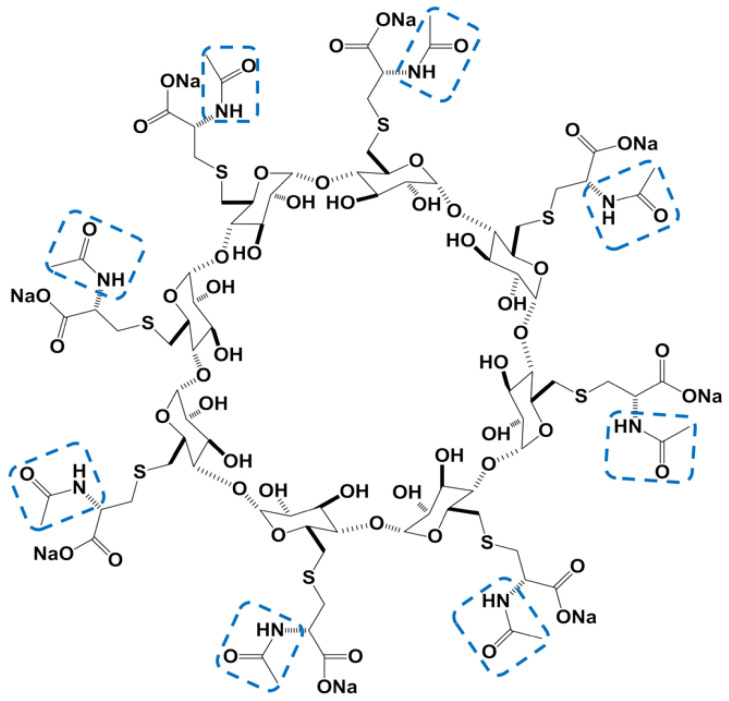
The chemical structure of adamgammadex.

**Figure 2 jcm-11-06951-f002:**
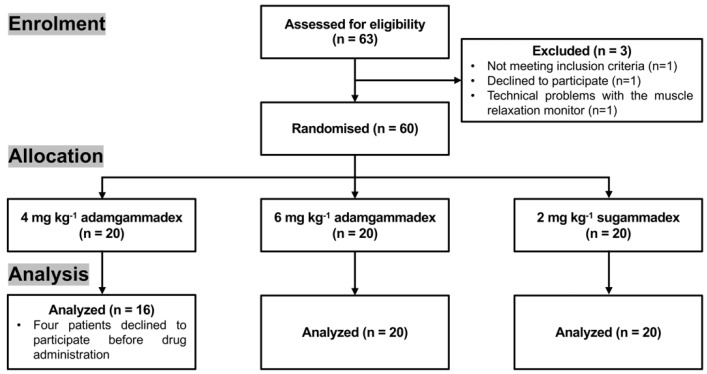
Study flow diagram.

**Figure 3 jcm-11-06951-f003:**
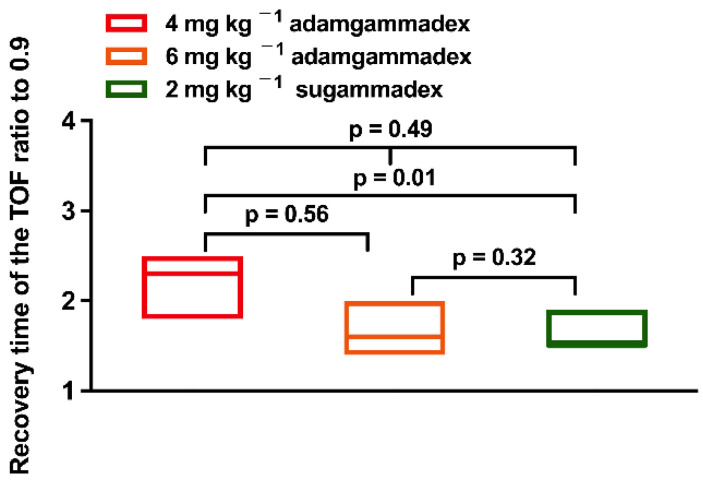
Time from start of administration of adamgammadex or sugammadex at the reappearance of T_2_ to the recovery of the TOF ratio to 0.9 (full analysis set).

**Figure 4 jcm-11-06951-f004:**
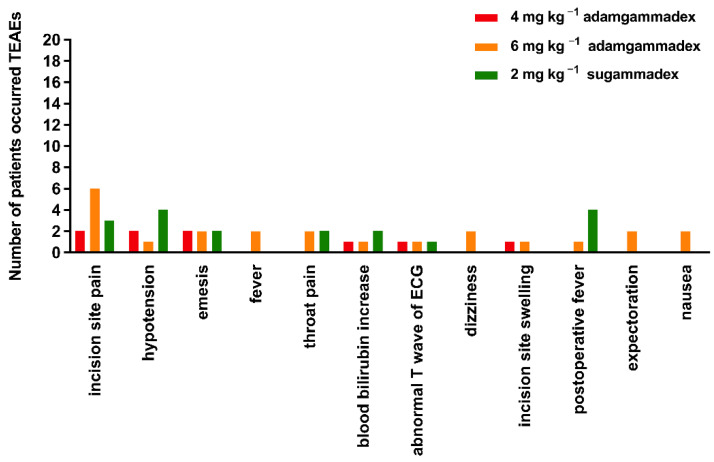
The number of treatment-emergent adverse events (TEAEs) in patients.

**Figure 5 jcm-11-06951-f005:**
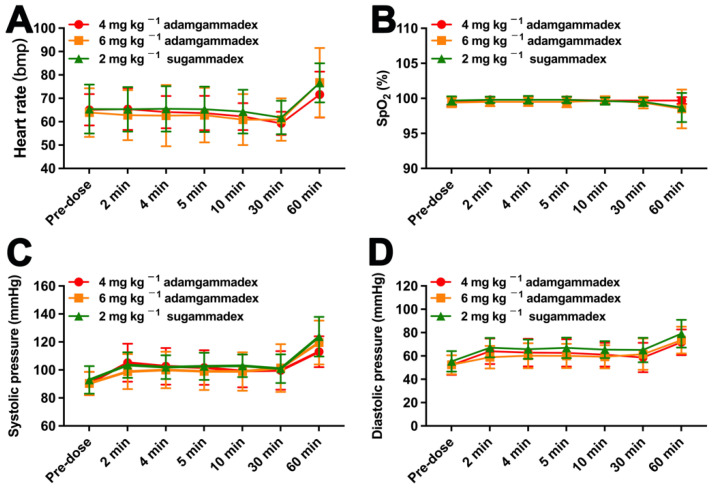
The basic vital signs, including heart rate (**A**), pulse-oximetry (SpO_2_, (**B**)), systolic pressure (**C**) and diastolic pressure (**D**), in patients after injected with adamgammadex or sugammadex.

**Figure 6 jcm-11-06951-f006:**
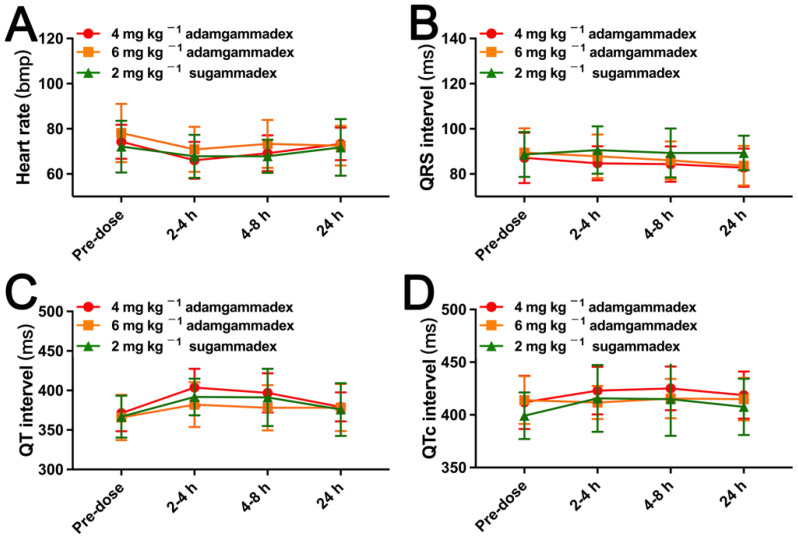
ECG variables, including heart rate (**A**), QRS interval (**B**), QT interval (**C**), and QTc interval (**D**), in patients after injection with adamgammadex or sugammadex.

**Table 1 jcm-11-06951-t001:** Subject demographics and baseline characteristics.

	Adamgammadex (4 mg kg^−1^, *n* = 20)	Adamgammadex (6 mg kg^−1^, *n* = 20)	Sugammadex (2 mg kg^−1^, *n* = 20)
sex (male), *n* (%)	7 (35%)	7 (35%)	10 (50%)
Age (years), min–max	21–57	23–60	20–57
Weight (kg), mean (SD)	62 (9)	63 (13)	63 (10)
Height (cm), mean (SD)	161 (7)	1635 (8)	165 (9)
BMI (kg/m2), mean (SD)	24.1 (2.7)	23.5 (3.4)	23.2 (2.7)
ASA class			
1	7	4	7
2	13	16	13
Surgical specialty			
Otolaryngology-Head and Neck Surgery	1 (5%)	3 (15%)	3 (15%)
Gynaecology	3 (15%)	2 (10%)	3 (15%)
Orthopaedics	0	0	1 (5.0%)
Oral and maxillofacial surgery	1 (5%)	1 (5%)	2 (10%)
Urology	0	1 (5%)	1 (5%)
General surgery	15 (75%)	13 (65%)	10 (50%)
Surgery duration (min), mean (SD)	58 (32)	81 (69)	68 (38)

**Table 2 jcm-11-06951-t002:** Adverse events by dose group: safety population set.

	Adamgammadex (4 mg kg^−1^, *n* = 16)		Adamgammadex (6 mg kg^−1^, *n* = 20)		Sugammadex (2 mg kg^−1^, *n* = 20)	
	Events	Patients	Events	Patients	Events	Patients
PTE, *n*	0	0	1	1	0	0
TEAEs, *n*	10	6	43	11	29	14
SAEs, *n*	0	0	0	0	0	0
drug-related AEs, *n* *	0	0	2 ^#^	2	0	0
drug-related SAEs, *n* *	0	0	0	0	0	0
anaphylactic reaction, *n*	0	0	0	0	0	0

* Considered to be possibly, probably, or definitely related to treatment. ^#^ one case of increased urine acetone bodies and one case of first-degree atrioventricular block. PTE, pretreatment event; TEAE, treatment emergent adverse events; AE, adverse event; SAE, serious adverse event.

**Table 3 jcm-11-06951-t003:** The coagulation function.

Parameters [Normal Range]	Pre-Dose			Post-Dose		
	Adamgammadex (4 mg kg^−1^)	Adamgammadex (6 mg kg^−1^)	Sugammadex (2 mg kg^−1^)	Adamgammadex (4 mg kg^−1^)	Adamgammadex (6 mg kg^−1^)	Sugammadex (2 mg kg^−1^)
PT (s), mean (SD) [9.6–12.8]	11.6 (1.2)	12.1 (1.5)	12.0 (1.5)	11.9 (1.4)	13.0 (1.5)	12.6 (1.6)
APTT (s), mean (SD) [24.8–33.8]	33.4 (6.4)	35.7 (5.7)	35.0 (6.6)	31.5 (6.4)	35.9 (5.6)	34.7 (6.7)
TT (s), mean (SD) [14.0–22.0]	16.3 (1.9)	16.1 (1.1)	17.0 (1.6)	15.5 (1.8)	15.9 (1.1)	16.1 (1.1)
Fib (g/L), mean (SD) [2.0–4.0]	3.1 (0.7)	3.4 (1.0)	2.8 (0.5)	3.4 (0.6)	3.7 (1.3)	3.3 (0.6)
INR, mean (SD) [0.9–1.2]	1.0 (0.1)	1.0 (0.1)	1.0 (0.1)	1.0 (0.1)	1.1 (0.1)	1.0 (0.1)

PT, prothrombin time; APTT, activated partial thromboplastin time; TT, thrombin time; Fib, fibrinogen; INR, international normalized ratio.

## Data Availability

All data will be made available upon request.

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
