# Peer review of "Comparison of the Efficacy and Safety of Adamgammadex with Sugammadex for Reversal of Rocuronium-Induced Neuromuscular Block: Results of a Phase II Clinical Trial"

_jcm, 2022, doi:10.3390/jcm11236951_

Round 1

Reviewer 1 Report

The authors addressed two interresting questions:

1. Efficacy of adamgammadex 4-6 mg/kg compared to sugammadex 2 mg/kg to reverse a moderate NMB.  The primary outcome was the reversal time from injection up to TOF ratio 0.9.  The results 2.3, 1.6, and 1.5 minutes respectively were considered not statistically different.  But only one p value was provided for 3 results: what about the comparison of adam 4 mg/kg compared to sug 2 mg/kg...2.3 and 1.5 seem quite different: please precise the statistical method of comparison, and why you did not compared each group to each other.  The conclusion stated that any dose of adam (4 or 6 mg/kg) was comparable to sug 2 mg/kg...which is not obvious when looking at the results: could you confirm this with an appropriate method ?

The secondary outcome was the reversal time up to TOF ratio 0.7 and 0.8: I really don't understand the usefullness of this analysis.  What did it add to the primary outcome ?  What is the clinical impact ?  Why didn't you discuss these outcomes ?  May I suggest to delete this part of the study ?

2. Safety of adamgammadex: the authors looked carefully at several side effects, and presented the occurence of some treatment emergent adverse events (TEAE).  This data collection was properly undergone but no link nor any possible explanation was established between the drugs investigated and the events reported.  Moreover, as the authors discussed, the sample size of this kind of study was insuffisant to state the incidence of the major adverse events attributed to sugammadex (particularly anaphylaxis).  The authors concluded well stating that a larger patient cohort should be evaluated.  This study was a first step.

Reviewer 2 Report

Reviewer’s comment

 This study compared the efficacy and safety of adamgammadex with sugammadex. From this study, adamgammadex is nearly equivalent to sugammadex in terms of the efficacy and safety. And the authors have shown that adamgammadex could be effective option for rocuronium-induced neuromuscular block reversal. This study is interesting and have clinical impact on patient safety. 

 Major comment.

1.      It has been widely known that anesthetics affect the sensitivity of neuromuscular blocker pharmacologic actions. For example, inhaled anesthetics may act synergistically with neuromuscular blockers. Also, deep anesthesia may reduce the required dose of neuromuscular blockers. Are there any differences in anesthetics (intravenous vs inhaled anesthetics) selection for each group in this study? Also, are there differences in depth of anesthesia between groups?

2.      From this study, it looks that low dose of adamgammadex group had a lower incidence of adverse effects (AEs) (mainly TEAEs), and the high dose group showed a higher number of AEs (mainly TEAEs). But there is no description of the difference of AES between the groups except the basic vital sign and ECG variables. Are there any differences of AEs between groups? In addition, it remains unclear as to whether any of the described AEs are due to surgical/anesthetic factors or the effects of drugs alone. For example, how can we confirm that emesis described as one of treatment emergent adverse events (TEAEs) is related to the use of neuromuscular reversal agents? 

3.      The authors classified the AEs of neuromuscular reversal agents into six categories. However, although the severity of AEs is clearly described in the manuscript, how these categories are defined is somewhat confusing. How did the authors classify the AEs of neuromuscular reversal agents?

4.      In terms of selecting dose of adamgammadex, the authors described that adamgammadex requires a higher dose than sugammadex to produce a similar effect. 4mg/kg or 6mg/kg agadammadex is equipotent to 2mg/kg sugammadex from a pharmacological point of view? If the dose is not equipotent, this may lead to the bias of efficacy and safety outcomes in this study?

Round 2

Reviewer 1 Report

Thank you for submitting a revised version of this manuscript.

I read your comment on my queries, but there is only minor adaptation to the manuscript.

1. Why do you analyse the recovery of TOFr 0.7 and 0.8 ?  Please comment this choice in the text (ex to follow the same method as it was done for sugammadex) or delete the data.  I think it is not useful neither for the interest of your study nor for the clinician.  The only valuable threshold for safe extubation is TOF ratio 0.9, for more than 3 decades.

2. As the -significant- difference between sug 2 mg/kg and adam 4 mg/kg reach a p value of 0.01 (Figure 3, A), you have to adapt all sentences in the results, discussion, conclusion and abstract.  It is not acceptable to write " the median recovery time of the TOFratio to 0.9 by the 4 and 6 mg kg-1 adamgammadex and 2 mg kg-1 sugammadex groups was not statistically significantly different" or "  there was no statistically significant difference between groups (p = 0.49)". 

The method of comparison between the 3 groups (or between each of them) was not specifically described and the goal of the study having being to compare the recovery profile of 2 doses of adam to sug 2 mg/kg, I don't understand why you provide such global result (no stat difference between 3 groups p 0.49) rather than the detailled analysis demonstrating that only the high dose of adam has no difference with sug (p 0.32) when the low dose was significantly different (p 0.01).

Please adapt all the text accordingly.  It is not enough to add a single sentence in the conclusion.
